# Integration of the Connectivity Map and Pathway Analysis to Predict Plant Extract’s Medicinal Properties—The Study Case of *Sarcopoterium spinosum* L.

**DOI:** 10.3390/plants11172195

**Published:** 2022-08-24

**Authors:** Valid Gahramanov, Moria Oz, Tzemach Aouizerat, Tovit Rosenzweig, Jonathan Gorelick, Elyashiv Drori, Mali Salmon-Divon, Michael Y. Sherman, Bat Chen R. Lubin

**Affiliations:** 1Department of Molecular Biology, Ariel University, Ariel 40700, Israel; 2Agriculture and Oenology Department, Eastern Regional R&D Center, Ariel 40700, Israel; 3Institute of Dental Sciences, Faculty of Dental Medicine, Hebrew University of Jerusalem, Jerusalem 9112001, Israel; 4Adelson School of Medicine, Ariel University, Ariel 40700, Israel; 5Judea Branch, Eastern Regional R&D Center, Kiryat Arba, Ariel 40700, Israel; 6Department of Chemical Engineering, Biotechnology and Materials, Ariel University, Ariel 40700, Israel

**Keywords:** plant medicine, RNA sequencing, SARS-COV-2, autophagy, signaling pathway

## Abstract

Medicinal properties of plants are usually identified based on knowledge of traditional medicine or using low-throughput screens for specific pharmacological activities. The former is very biased since it requires prior knowledge of plants’ properties, while the latter depends on a specific screening system and will miss medicinal activities not covered by the screen. We sought to enrich our understanding of the biological activities of *Sarcopoterium spinosum* L. root extract based on transcriptome changes to uncover a plurality of possible pharmacological effects without the need for prior knowledge or functional screening. We integrated Gene Set Enrichment Analysis of the RNAseq data to identify pathways affected by the treatment of cells with the extract and perturbational signatures in the CMAP database to enhance the validity of the results. Activities of signaling pathways were measured using immunoblotting with phospho-specific antibodies. Mitochondrial membrane potential was assessed using JC-1 staining. SARS-CoV-2-induced cell killing was assessed in Vero E6 and A549 cells using an MTT assay. Here, we identified transcriptome changes following exposure of cultured cells to the medicinal plant *Sarcopoterium spinosum* L. root extract. By integrating algorithms of GSEA and CMAP, we confirmed known anti-cancer activities of the extract and predicted novel biological effects on oxidative phosphorylation and interferon pathways. Experimental validation of these pathways uncovered strong activation of autophagy, including mitophagy, and excellent protection from SARS-CoV-2 infection. Our study shows that gene expression analysis alone is insufficient for predicting biological effects since some of the changes reflect compensatory effects, and additional biochemical tests provide necessary corrections. This study defines the advantages and limitations of transcriptome analysis in predicting the biological and medicinal effects of the *Sarcopoterium spinosum* L. extract. Such analysis could be used as a general approach for predicting the medicinal properties of plants.

## 1. Introduction

Understanding the physiological effects of medicinal plant extracts is a vast area of biomedical research [1]. Despite the widespread use of herbs, knowledge about their biological activity is still limited. The herbal extracts are composed of thousands of different substances that together can have synergistic or antagonistic effects [2,3]. For example, such complex effects were found in studies of *Hydrastis canadensis* [2,4] or *Artemisia annua* [5], demonstrating the complexity of the composition of the extracts. The complexity of herbal medicine composition could create difficulties in isolating specific compounds of interest. On the other hand, as most diseases are multifactorial and involve several physiological mechanisms, whole or partially fractionated plant extracts could be uniquely suitable for treating complex medical conditions.

A standard approach toward finding a medicinal plant activity is via conducting biological screens. For example, the MTT test has been used to assess the anti-cancer activity of *Inula Viscose* or *Retama monosperma* L extracts in human cervical cancer cells [6]. This approach is limited as it is highly focused on a specific activity of the plant extract. Accordingly, one needs to have prior ethnobotanical knowledge of the use of a particular plant in the treatment of a studied disease; see, for example, [7,8,9]. If such ethnobotanical knowledge is missing, the successful search has to include a widescreen of plants, using dozens or hundreds of extracts [10].

A growing body of literature describes the use of transcriptomic analysis in studies of disease development and the identification of new biomarkers and drug mechanisms of action [11,12,13]. For example, in one of the previous studies via RNA-seq analysis, researchers identified a different gene signature that can be useful for predicting the prognosis in neuroblastoma [14]. In another example, the transcriptomic analysis uncovered a list of genes important for the pathogenesis of autism spectrum disorder [15]. Recently, a new approach was developed to simplify analysis of the transcriptome by utilizing a small set of RNA species, termed the Connectivity Map (CMAP) database. The innovative advantage of this approach is that a very large database has been built, incorporating transcriptomic signatures of effects of tens of thousands of drugs and bioactive compounds on several cell lines [16]. These gene set signatures provide sufficient information to establish new drug targets and their mechanisms of action [17]. These and other studies also showed that transcriptomic analysis could be useful in targeted drug development. Several works in traditional Chinese medicine (TCM) have used the CMAP set analysis to study the mechanisms of the TCM formula. These studies have tested 102 known plant-derived compounds [18,19] and the whole extract [20,21] first by limited gene expression microarray and then by CMAP, resulting in gene expression profiles that serve as a template for general TCM research.

Here, we used a different bioinformatic approach by integrating CMAP datasets with GSEA to analyze the potential biological activities of an extract from the roots of *Sarcopoterium spinosum* L., a traditional Mediterranean plant [22]. Many ethnopharmacological surveys reported the use of *Sarcopoterium spinosum* L. root extract (SSRE) for the treatment of diabetes [22,23,24,25,26], cancer therapy [26,27,28], and intestinal diseases [25,26]. Additionally, a positive role of the extract in inflammatory conditions has been investigated [22,26,29,30]. Chemical analysis of *Sarcopoterium spinosum* L. root extract demonstrated the existence of pentacyclic terpenoids such as tormentic and ursolic acids [31]. In addition, catechins, epicatechins, and dimeric forms of catechins were identified, as well as proanthocyanidines and β-sitosterol [31,32,33]. However, the extract is composed of many additional compounds, which have not been identified so far. This investigation not only confirms previously known medicinal properties of the extract but also predicts previously unknown activities that we validated. Such analysis could be used more widely in predicting novel medicinal properties of plants.

## 2. Results

### 2.1. Evaluating the Cytotoxicity Effect of Root Extract from Sarcopoterium spinosum L.

Here, we sought to search for additional medicinal properties of *Sarcopoterium spinosum* L. (*SSRE*), which has been studied for its anti-diabetic and anti-cancer activities. To avoid extensive apoptotic signature in the transcriptome analysis that could mask other important gene expression changes, we first identified subtoxic concentrations and incubation time with the extract. Accordingly, *S. spinosum* extract was added to a series of cell lines, including A20 mouse lymphoma, HL-60 human promyelocytic leukemia, NB-4 human acute promyelocytic leukemia, and A549 human lung cancer cells. Cell death was measured by the XTT assay after 24 h of incubation (Figure 1A). For further experiments, we decided to use the most resistant cell line, A549 treated with 200 µg/mL of the extract for 6 h, thus preceding possible toxicity effects.

### 2.2. RNAseq and Pathway Analysis

To identify the effects of the extract on the gene expression pattern of A549 cells, RNA was isolated from the cells, and RNAseq was performed. A total of 1331 differentially expressed genes (fold change of 2, and FDR < 0.05) between treated and non-treated samples were identified. Among these genes, 601 were up-regulated, and 730 were down-regulated. The overall landscape of gene expression is shown as Volcano Plot in Appendix A, and the list of differentially expressed genes with their expression values is given in Appendix A. Top differentially expressed genes are visualized by heatmap in Appendix A.

To get an intuition about pathways that are potentially activated or inhibited by the extract, we analyzed RNAseq results using the protein-protein interactions STRING database (https://string-db.org, accessed on 18 October 2021) (Figure 1C). Further, GSEA was performed to determine whether pathways predicted from the STRING database are statistically enriched between two biological states (treatment and control). GSEA was employed against the hallmark gene-set signature. Hub genes detected in STRING analysis were found by GSEA to be associated with several critical pathways, including downregulation of WNT/Beta-catenin pathway, TGF-beta signaling, c-Myc signaling, G2/M checkpoint, Spliceosome pathway, and upregulation of Oxidative Phosphorylation and Interferon signaling (Appendix A).

### 2.3. Comparative Computational Pathway Analysis of the SSRE Extract Effect on Various Cell Lines

A potential problem with the proposed approach is that different cell lines may respond differently to the same plant extract. Therefore, treatment of multiple lines may be needed to obtain a reasonably comprehensive understanding of the potential biological and medicinal effects of the extract. Therefore, we compared the pathway analysis of the transcriptome changes in human lung cancer cells A549, and in dramatically different cells, mouse 3T3 fibroblasts differentiated to adipocytes (Appendix A). As shown in Figure 1B, the pathways that are either activated or suppressed by the extract treatment are almost the same in both cell lines, even though we could see differences in genes that are enriched in the biological processes (e.g., in the oxidative phosphorylation pathway). These data suggest that regardless of the origin of a cell line, one can derive potential biological activities of the chosen extract by using transcriptomic analysis, and therefore there seems to be no need to use multiple cell lines.

### 2.4. Integrating CMAP into GSEA Analysis of the SSRE Transcriptome Signature

To further enhance our understanding of the potential biological activities of the extract, we have utilized information from the connectivity map (CMAP). Originally, CMAP was built to compare drug-specific gene expression profiles using a reference database. The algorithm used to build the CMAP is entirely different from that used in GSEA, and therefore integrating the CMAP will enhance the validity of GSEA analysis of the transcriptomic changes (Figure 2A). First, we compared the transcriptome signature of SSRE treatment with drug-induced signatures in CMAP. Ten top hits with the highest scores were compounds that belong to three pathways, including RTKs, NFkB pathway, and GPCRs (Figure 2B). By extracting the gene expression data of these top hit compounds in the CMAP database, we performed pathway analysis and compared the results of enriched pathways of compounds to our dataset. Interestingly, there was a clear overlapping of certain pathways, including spliceosome, cell cycle, oxidative-phosphorylation, and autophagy pathways that were commonly found in all three groups of compounds and our RNA-seq analysis (Figure 2C) (Appendix A). Since the CMAP database contains approximately a million gene expression signatures from the treatment of a variety of cell types with perturbagens, thus overlap of these pathways empowers the idea of finding the actual biological activities of the medicinal plant extract via transcriptomic analysis.

### 2.5. Validating the Pathways and Predicting Anti-Cancer Effects of SSRE

Based on the integrated bioinformatics analysis, we focused on overlapping pathways that showed up both in GSEA and CMAP analysis. Our first question was whether the transcriptome analysis could predict known anti-cancer activities of *SSRE*, and thus we focused on pathways relevant to tumor growth and cancer progression. To validate the effects of the extract on cancer-related pathways, inhibition of the WNT/Beta-catenin pathway and TGF-beta signaling was tested by immunoblotting with main intracellular signaling elements anti-phospho-β-catenin and anti-SMAD2 antibodies, respectively. The phospho-β-catenin level was reduced upon exposure to extract, reflecting its degradation (Figure 3A). Similarly, the level of SMAD2 was reduced after the administration of the extract (Figure 3A). Thus, both upstream signaling elements of the pathways and downstream gene expression changes were downregulated upon exposure of cells to the extract. We also validated the effects of the extract on the expression of the centrosome (part of the G2/M pathway) and spliceosome genes by RT-PCR (Figure 3B).

One of the crucial genes involved in cell cycle progression is c-Myc. The altering expression of c-Myc is indicative of cell proliferation as well as the progression through the G1 phase [34,35]. Downregulation of c-Myc can lead to G1 arrest, while inhibition of the G2/M checkpoint to the G2/M arrest [35,36]. Therefore, the prediction based on the pathway analysis was that the extract might cause suppression of both the G1 and G2 phases of the cell cycle. Indeed, FACS analysis of naïve cells and cells treated with the extract indicated depression in S-phase and an increase in G1 cell populations (Figure 3C).

Besides WNT/Beta-catenin, TGF-beta, and c-Myc signaling, as well as G2/M checkpoint, downregulation of the spliceosome pathway also suggests an anti-cancer activity since splicing is significantly affected by cancer transformation [37,38], and inhibitors of splicing have been developed for cancer treatment [39]. Altogether, downregulation of these pathways indicates that previously demonstrated anti-cancer activities of the extract [22,26] could indeed be predicted from the RNAseq analysis.

Alhough there is an extensive literature on the anti-diabetic activity of *SSRE*, our analysis did not show any indications of such an activity. In other words, we did not observe any transcriptional effects related to the metabolic changes that occur in diabetes. This lack of effect simply reflects the fact that the anti-diabetic activity of the extract is unrelated to transcription changes, see Discussion. Therefore, overall, medicinal activities related to the effects of an extract on transcriptome can be predicted based on the transcriptome analysis, but transcription-independent effects could be missed.

### 2.6. Previously Unknown Anti-Viral and Autophagy-Stimulating Effects of Sarcopoterium spinosum L. Extract

Among the top highly enriched pathways, we observed strong upregulation of the Interferon pathway and Oxidative Phosphorylation (Appendix A). Further, we validated the upregulation of a set of interferon-γ-induced genes and mitochondrial genes in response to incubation with the extract by RT-PCR (Figure 4A).

Since the Oxidative Phosphorylation pathway was upregulated by the extract according to the transcriptome analysis (both nuclear-encoded and mitochondria-encoded genes, see Appendix A), we used this information to predict the novel biological activities of *SSRE*. If the upregulation is associated with higher mitochondrial content and improvement of oxidative phosphorylation, treatment with the extract could be very useful in alleviating diseases associated with the accumulation of defective mitochondria, like neurodegenerative disorders such as Parkinsonism or Friedreich ataxia [40,41,42,43]. Accordingly, we sought to test whether treatment with the extract could enhance the production of mitochondria in cells by utilizing a fluorescent probe MitoTracker. A549 cells were treated with the extract for 6 h and stained with Mitotracker according to the manufacturer’s protocol. Images were taken and analyzed using the Hermes Imaging system. Surprisingly, and against the expectations, instead of the higher number of mitochondria in treated cells, we observed a significant and time-dependent decrease in the number of mitochondria (Figure 5A,B).

Upon analysis of these data, we realized that the extract could damage mitochondria and thus make them more susceptible to selective autophagy or alternatively, the extract could cause overall activation of the mitophagic pathway. To distinguish between these possibilities, we first assessed the membrane potential of mitochondria (ΔΨ) JC-1 staining since mitochondrial damage usually leads to the collapse of ΔΨ [44]. In healthy cells, JC-1 accumulates in the energized mitochondria and forms red fluorescent J-aggregates. By contrast, upon the collapse of ΔΨ, JC-1 forms green, fluorescent J-aggregates. As shown in Figure 6A, incubation with the uncoupler FCCP that causes the collapse of ΔΨ leads to the shift of the JC-1 fluorescence from red to green. In contrast, incubation with the extract did not cause such a shift, indicating healthy ΔΨ. Therefore, the decrease in the number of mitochondria upon treatment with the extract was not associated with mitochondrial damage.

To test the possibility that the extract over-activates the autophagic pathway, we inhibited lysosomal degradation by hydroxychloroquine and measured the number of mitochondria before and after administration of the extract. Indeed, inhibition of autophagy significantly reversed the effect of the extract on the number of mitochondria (Figure 6B), indicating the autophagy-dependent mitochondrial degradation.

To test if the extract activates the overall autophagic pathway, a Western blot analysis of LC3 protein processing was done. Indeed, treatment of A549 cells with the extract led to an increase in LC3B-II levels (Figure 7B). Furthermore, we observed a decrease in the level of an autophagic receptor p62 that recognizes autophagic targets and recruits them to the autophagic vacuoles (Figure 7A). Such a decrease is expected upon activation of the autophagic flux due to degradation of p62 [45]. Therefore, treatment of cells with the *SSE* activates the overall autophagic pathway and facilitates the autophagic degradation of mitochondria and probably other autophagic targets. Such feature of the *SSRE* could be very useful for the treatment of cancer (since autophagic activators have been used for cancer treatment) [46] and alleviation of various neurodegenerative diseases associated with defects of mitochondrial degradation, like Parkinsonism [47].

Strong upregulation of the γ-interferon pathway seen in our computational analysis predicted that the extract could have anti-viral activities. To evaluate possible anti-viral effects of *SSRE*, we tested whether the extract could protect cells from toxicity caused by SARS-CoV-2 (the virus causing COVID19) infection using a standard approach for testing anti-COVID activities of drugs. VERO E6 or A549-HA-FLAG cells were pretreated with extract for 4 h followed by a 1 h exposure to SARS-CoV-2 infection. After 48 h, cell death caused by the viral infection was evaluated using an MTT assay. Figure 4B demonstrates that while viral infection led to the death of 50–60% of cells, treatment with the extract mitigated this toxicity almost completely. Therefore, an activity of the *SSRE* against a broad spectrum of viruses could be predicted based on the transcriptome analysis and was validated in vitro in a standard test for COVID19 propagation and toxicity.

These two examples with the interferon pathway and oxidative phosphorylation pathway indicate that integration of GSEA and CMAP in the transcriptome analysis could be used not only for confirming known activities but also for predicting novel medicinal activities of plant extracts.

## 3. Discussion

Transcriptome analysis has been extensively used to predict the biological effects of various biologically active compounds [11]. With plant extracts, however, the situation could be problematic because of the chemical complexity of the extracts, which could contain hundreds of biologically active ingredients [48,49]. The goal of this research was to get a generalized idea of the biological effects of the *S. spinosum* extract through transcriptome analysis. Such an analysis could hint toward the potential medical effects of the plant, which requires linking biological pathways to specific diseases. Prior studies have attempted to analyze the effects of certain plant extracts on the transcriptome of target cells or tissues [24,26]. However, predicting biological or pharmacological activities of plant extracts based on these data has not been done.

An important advance in this work was the integration of GSEA and CMAP analysis. Indeed, usually use of CMAP is limited to finding drugs with similar effects on the transcriptome, suggesting similar activities. Here, we utilized CMAP differently by extracting gene sets activated or inactivated by drugs that scored between 100 and 95 and performing parallel pathway analysis. Integrating these data with GSEA analysis significantly enhanced the confidence in the results. Of note, CMAP analysis revealed three classes of compounds, including inhibitors of RTKs, GPCRs, and NF-kB pathway, which normally would be suggested to show similar biological activities of these classes of inhibitors to the *SSRE*. However, upon extraction of the corresponding gene sets and pathway analysis, it became clear that all three distinct classes of inhibitors showed a similar set of pathways, which could represent either secondary or off-target effects. These pathways overlap with pathways activated by *SSRE*. Therefore, such non-canonical use of CMAP may be very informative in the analysis of potential biological effects of plant extracts.

There have been significant efforts to build databases that integrate information about biological pathways and various diseases to understand the relationships between drugs, genes, and diseases [50,51,52]. A public database, MalaCard could be quite helpful in making predictions about the medicinal properties of plant extracts based on transcriptome analysis [53]. One can input pathways or individual genes, and the database provides a list of diseases associated with this list. A drawback of MalaCard is that it is somewhat unbalanced (https://www.malacards.org/, accessed on 24 November 2021) since it is based on published articles, and the highest scores are usually for various types of cancer while other diseases are underrepresented.

High similarities between two entirely different cell lines in pathways affected by *SSRE* suggest that in the predicted pharmacological effect of an extract via transcriptome analysis, one can use a few or maybe even one cell line. This observation significantly supports the feasibility of our approach. Overall, this integrated analysis of GSEA and CMAP datasets will make it feasible to develop an idea about the potential medicinal properties of complex plant extracts.

Since WNT/Beta-catenin pathway, TGF-beta signaling, c-MYC signaling, and G2/M checkpoint are involved in various aspects of cancer development [54,55,56,57], downregulation of all these pathways upon exposure to *SSRE* can be used to predict anti-cancer properties of the extract. Indeed, downregulation or inhibition of these pathways was shown to suppress cancer development, while their upregulation stimulated cancer [57,58,59]. Importantly, this RNAseq analysis can significantly enrich our understanding of the mechanism of anticancer activity of the extract. Indeed, previous publications only reported anti-proliferative and cytotoxic activities of SSRE and demonstrated that tormentic acid is the bioactive component. [27,60]. While downregulation of c-MYC and G2/M checkpoint genes explains the antiproliferative activity, downregulation of TGF-beta and WNT/Beta-catenin pathways suggest the suppression of cancer cells stemness and metastasis. In addition to cancer, these pathways are involved in the development of other disorders, as well. TGF-beta signaling is implicated in pathological skin disorders, including excessive scarring and chronic wounds [61,62]. Several reports have shown that inhibition of the TGF-beta signaling pathway may be a good strategy for wound healing and reduction of pathological scarring [62]. In other words, identifying signaling pathways affected by an extract may suggest beneficial effects of the extract for the treatment of multiple disorders.

Our distinct finding was strong upregulation of interferons α and γ pathways, suggesting that the extract could have anti-viral activities. Indeed, interferons elicit broad anti-viral responses, interfering directly with both viral entry and replication as well as stimulating an immune response through activation of macrophages and NK cells [63,64,65,66,67]. Both interferon-α and interferon-γ were reported to reduce viral titers of SARS-CoV-2 [68,69], and it is entirely plausible that these interferons are involved in SSRE’s anti-viral activity. Anti-viral activity of SSRE was not demonstrated before thus the bioactive chemicals mediating this activity are completely unknown. Interestingly, molecular docking studies suggest that tormentic acid, a component of *SSRE*, regulates JAK1 and STAT3, which are involved in interferon signaling [70]. However, the anti-viral activity of this compound should be validated in wet studies. In addition to tormentic acid, anti-viral activity was attributed to proanthocyanidins [71], an additional composite of *SSRE*. Thus, a bio-guided fractionation approach should be utilized in order to isolate and identify the anti-viral molecules in *SSRE* [72].

There have been studies to use herbs and mushrooms against COVID-19, including several trials with herbal and mushroom mixtures [73,74]. Results of these trials are still under investigation, and there is little mechanistic information, including information on the involvement of interferons in the effects of these herbs on the propagation of the virus. Moreover, other reports regarding the effect of natural compounds extracted from medicinal plants have shown. Lin et al. (2014) summarized the list of various medicinal plants (*Bupleurum* spp., *Heteromorpha* spp., and *Scrophularia*) that show antiviral activity against human coronaviruses by affecting viral attachment and cell penetration of the virus [75]. In addition, several traditional Chinese medicines (TCM) prescriptions, such as Pudilan (a combination of a four-herb prescription), have been utilized for SARS-CoV-2 treatment [76]. In this research, by using network pharmacology analysis, the authors, based on the target gene analysis, computationally demonstrated possible effects of the extracts on viral penetration and suppression of the cytokine storm caused by COVID-19. However, these predictions have not been validated in wet-lab experiments. To summarize, analysis of natural agents against COVID-19 mostly does not provide sufficient mechanistic information, and further investigation on possible biological pathways related to the disease is required.

A well-known activity of the *SSRE* is the alleviation of Type 2 Diabetes [24,26]. These effects have been extensively investigated in animal models and cell culture, showing that the extract dramatically enhances insulin signaling [77]. However, in our experiment, we could not see any signatures of these effects in the transcriptome analysis and, therefore could not predict these biological and medicinal effects. However, such a lack of predictive ability appears to result from the lack of association of the anti-diabetic effects of the extract with transcription changes. Indeed, the extract directly influences insulin signaling [78]. Its effects on the signaling pathway are seen within minutes of administration and therefore are transcription-independent [78]. Thus, our approach to predicting the biological and medical effects of plant extracts based on the transcriptome analysis has the limitation that it may not be able to uncover transcription-independent effects.

Another important effect was the induction of autophagy. Indeed, treatment with extract led to a decrease in p62 and an increase in LC3 processing, which reflects activation of the autophagic flux [79,80]. Therefore, treatment of cells with the *SSRE* activates the overall autophagic pathway and facilitates the autophagic degradation of mitochondria and probably other autophagic targets, e.g., protein aggregates. Such features of the *SSRE* could be very useful for the treatment of cancer and alleviation of various neurodegenerative diseases related to defects in mitochondrial degradation, like Parkinsonism [40,41,47].

To our surprise, treatment with the extract, while upregulating mitochondrial genes, reduced the mitochondrial compartment in cells. Therefore, upregulation of the mitochondrial genes seen in the extract-treated cells reflected a compensatory induction of these genes. Accordingly, upon analysis of the RNAseq results, one should be aware that changes in gene expression could reflect compensatory responses rather than genuine activation of the pathway. Thus validations of the pathway activity are critical.

## 4. Materials and Methods

### 4.1. Plant Collection

Roots of *Sarcopoterium spinosum* L. were collected in December 2018 and uprooted from wild-growing plants at the edges of the Ariel University campus in accord with the laws of Israel’s authorities for biodiversity. A voucher specimen of the plant was deposited in the Israel National Herbarium at the Hebrew University of Jerusalem (No. HUJ 102531).

### 4.2. Aqueous Extract

In accordance with the data published in ethnobotanical surveys [22,26,81], roots of *Sarcopoterium spinosum* L. were washed and boiled for 30 min in a ratio of 1 gr:10 mL volume of water. After cooling to room temperature, the extract was filtered (Whatman No. 4). The extract was lyophilized and kept at −20 °C.

### 4.3. Cell Lines

The lung carcinoma A549 cell line was grown in DMEM complete medium supplemented with 10% heat-inactivated fetal calf serum and 1% Penicillin-Streptomycin mixture. A20 mouse lymphoma cells, HL-60 human promyelocytic leukemia cells, NB4 human promyelocytic cells, and K562 chronic myelogenous leukemia cell lines were grown in suspension in RPMI supplemented with 10% heat-inactivated fetal calf serum and 1% Penicillin-Streptomycin mixture. 3T3-L1 pre-adipocytes were cultured and induced to differentiate. 3T3-L1 adipocytes were used for experiments 14 days after the initiation of differentiation when 80–90% of cells exhibited adipocyte morphology. Cultures were maintained at 37 °C in 5% CO_2_.

### 4.4. Cytotoxicity Assay

Cytotoxicity of the plant extracts was determined using the XTT Assay [82] (Biological Industries, Israel). 1.5 × 10^4^ A549 cells and 8 × 10^5^ cells of A20, HL-60, NB-4, and K562 were seeded in 96-well microplates and allowed to recover. The next day, cells were treated with extract at concentrations ranging from 50 to 600 μg/mL, in triplicate for 24 h incubation. 50 μL XTT (5 mg/mL) was added to each well. After 2 h of incubation, absorbance was determined using a spectrophotometer at 450 nm. To measure non-specific reading, a 630–690 nm wavelength was used.

### 4.5. RNA Extraction and Library Construction

A549 cells were seeded into 6-well plates at a density of 4.5 × 10^5^ cells/mL. The cells were treated with 200 μg/mL *SSRE* for 6 h. Total RNA was extracted using GENEZOL TRIRNA PURE KIT (Geneaid, Taiwan) according to the manufacturer’s protocol. RNA concentration was quantified using Qubit^®^ dsDNA HS Assay Kit (Thermo Scientific, Wilmington, DE, USA).

Standard library construction was performed using the TruSeq Stranded mRNA Library Prep kit for Illumina. For each biological replicate, 100 ng of mRNA was used for each library.

### 4.6. RNA Sequencing, Quality Control, Sequence Alignment, and Gene Counts

RNAseq libraries of untreated and treated samples were prepared using Truseq Stranded mRNA Prep Kit (Illumina) according to the manufacturer’s protocol. RNAseq libraries were sequenced by Illumina Hiseq 2500 platform. The generated raw reads underwent quality control checking procedures that included the removal of adapter sequences and low-quality reads. Evaluation of the quality of reads of each sample based on Phred quality score [83] has been done by using FASTQC [84] (version 1.1.9, http://www.bioinformatics.babraham.ac.uk/projects/fastqc/, accessed on 20 August 2022) software. Further, obtained high-quality reads were aligned against the human genome (build hg38) using the fast and sensitive alignment program HISAT2 [85] (http://daehwankimlab.github.io/hisat2/, accessed on 20 August 2022) with default parameters. The number of reads per gene was measured using featureCounts software [86] (version 1.22.2, the Australian National Health and Medical Research Council (NHMRC), Victorian State Government Operational Infrastructure Support and Australian Government NHMRC IRIIS).

### 4.7. Normalization and Differential Expression Analysis

To make accurate comparisons of gene expression between untreated and treated samples, TMM normalization of counts has been done by edgeR R package [87]. For data transformations for RNA-seq differential expression analysis, we utilized the voom transformation [88]. Limma package [89] was used to generate linear models for the detection of differentially expressed genes. Correction for multiple comparisons was done using the Benjamini–Hochberg (BH) correction method and genes having an FDR < 0.05 and fold change > 2 were considered differentially expressed. Further, heatmaps of differentially expressed genes were generated using the heatmap function in R programming language using Euclidian as the distance measure and Ward.D2 as the linkage method.

### 4.8. Functional Analysis of RNA-seq (GSEA, STRING, CMAP)

Search Tool for the Retrieval Interacting Genes [90] (STRING) (https://string-db.org, accessed on 20 August 2022) is an online software for analyzing interactions of genes and proteins. Protein–protein interaction (PPI) of the differentially expressed genes was constructed from the STRING database. Constructed interaction patterns were visualized by Cytoscape [91] (software version 3.8.2). For the construction of PPI networks, active interaction sources, including text mining, experiments, databases, and co-expression, as well as species limited to “Homo sapiens” and an interaction score > 0.4 were applied.

The entire list of genes was ranked according to fold change and used as input to Gene Set Enrichment Analysis [92] (GSEA). GSEA was employed against the hallmark gene-set signature. The hallmark gene sets were obtained from the Molecular Signature Database v7.2. (http://www.gsea-msigdb.org/gsea/msigdb/index.jsp, accessed on 20 August 2022) was used in the analysis. We have taken into consideration gene sets with a *p*-value < 0.05 and FDR cutoff of 25%, which is appropriate in GSEA analysis due to the relatively small number of gene sets being analyzed (50) [92]. Transcriptome alterations, including upregulated and downregulated DEGs, were compared with gene signatures in CMAP database [93]

### 4.9. Cell Cycle Analysis

A549 cells were seeded into 24-well plates at a density of 1 × 10^5^ cells/mL. The cells were treated with 200 µg/mL of the extract for 24 h. The cells were harvested using 0.1% trypsin-EDTA. Cells were then fixed and stained using a propidium iodide flow cytometry kit (Abcam) according to the manufacturer’s protocol. DNA content of the cells was analyzed using a flow cytometer with a Cytoflex FACS Calibur cell analyzer.

### 4.10. cDNA Synthesis

The complementary DNA (cDNA) was synthesized using 50 ng/μL RNA of each sample. The reverse transcription (RT) reaction was performed according to high-capacity cDNA reverse transcription kit instructions (AppliedBiosystems, Inc.).

### 4.11. Real-Time PCR

cDNA was analyzed with real-time polymerase chain reaction (PCR) using the Sequence Detection System (ABIPrism7900; AppliedBiosystems, Inc., Foster City, CA, USA). cDNA input levels were normalized against human beta-actin (ACTB). Reactions were performed in a 12 μL volume containing 4 μL cDNA,0.3 μL each of forward and reverse primers, 6 µL buffer included in the MasterMix, and 1.4 µL of DNAse/RNase-free water (SYBRRGreenI; AppliedBiosystems, Inc.). PCR cycling conditions were as follows: initial denaturation at 50 °C for 2 min; followed by denaturation at 95 °C for 2 min; followed by 40 cycles of denaturation at 95 °C for 15 s; and annealing and extension at 60 °C for 1 min. Triplicate was performed for each gene to minimize individual tube variability, and an average was taken for each sample.

The results were quantified by a comparative Ct method, also known as the 2ΔCt method [94].
**Gene Symbols****Forward****Reverse*****IFIT2***gacacggttaaagtgtggaggtccagacggtagcttgctatt***IFIT1***tgagatgtcactttacatgggtgtattcccacactgtatttgg***CCP110***agacgcagtctgagaggtagtcagtgtttgcctgtcaactgg***CEP152***ggagtggcagtctaagctggtcactggtggttacttggtca***NDUFB11***cgtccgctgggaatctagcacggggtccttgtcataacca***COX5B***tgtgaagaggacaataccagcgccagcttgtaatgggctccac***COX6B1***ctacaagaccgccccttttgatttagcggtcattgccttctg***SRSF11***caggtgactaatgtctccccggcagttcgtcgatcttgcct***PNN***gaatgacgtgaggcccatccactctgtttggctgggggtcct

### 4.12. Analysis of Anti-Viral Activity

Cell lines VERO E6 (ATCC^®^ CRL-1586) and Human Lung Carcinoma Cells (A549) Expressing Human Angiotensin-Converting Enzyme 2 (HA-FLAG) (NR-53522) as well as SARS-CoV-2 (Isolate USA-WA1/2020) were obtained from BEI Resources. All experiments were carried out in a Biosafety level 3 laboratory and conducted under appropriate conditions. Vero E6 cells were cultured in Minimum Essential Media (MEM) supplemented with 2% fetal bovine serum, penicillin/streptomycin (100 μg/mL), and L-glutamine (2 mM). A549 cells were cultured in Dulbecco’s Modified Eagles Medium (DMEM) supplemented with 10% Fetal Bovine Serum, 2 mM L-Glutamine, and 1 μg/mL puromycin. All cells were cultured at 37 °C under 5% CO_2_.

Cells were seeded in 96-well flat-bottom microplates at 5 × 10^4^ cells/mL and incubated overnight to reach sub-confluence. On the next day, cells were pretreated with *S. spinosum* extract (100 µg/m) for 4 h, followed by infection with SARS-CoV-2 at various multiplicities of infections (MOI). After 48 h, viral-induced cytotoxicity was evaluated using MTT (Invitrogen) based on the manufacturer’s protocol.

### 4.13. Analysis of Mitochondrial Activity

A549 cells were seeded into 96 well clear bottom black plates at a density of 12 × 10^5^ cells/mL. The cells were treated with 200 μg/mL extract for 6 h. After incubation, the cells were stained with MitoTracker Red (Invitrogen) according to the manufacturer’s instructions. Representative images were obtained via WiScan^®^ Hermes High Content Imaging System (IDEA Bio-Medical Ltd., Rehovot, Israel) at room temperature using a 20× objective. The data obtained from the microscope were analyzed with Athena software.

### 4.14. Analysis of Mitochondrial Membrane Potential (ΔΨm)

A549 cells were seeded as previously described. The cells were treated with 200 μg/mL extract for 6 h and stained with JC-1 (Abcam) according to the manufacturer’s instructions. Representative images were obtained under a Zeiss confocal microscope equipped with a laser diode 488 nm and 555 nm.

### 4.15. Autophagy Study

A549 cells were seeded into 96 well black plates and clear bottom at a density of 12 × 10^5^ cells/mL and then incubated with 60 µM hydroxychloroquine (Sigma-Aldrich, Rehovot, Israel), and 200 μg/mL extract simultaneously or extract only for 6 h. Cells were stained with JC-1 according to the manufacturer’s instructions. Absorbance in the wells was measured at 530 nm and 590 nm. All experiments were performed in triplicate on three separate occasions. Data are presented as mean ± S.D.

### 4.16. Western Blot Analysis

A549 cells in 35 mm or 60 mm dish were lysed with lysis buffer: 40 mM HEPES, pH 7.5; 50 mM KCl; 1% Triton X-100; 2 mM dithiothreitol; 1 mM Na3VO4; 50 mM β-glycerophosphate; 50 mM NaF; 5 mM EDTA; 5 mM EGTA; and supplemented with Proteasome inhibitor Cocktail (Sigma) and PMSF before use. Samples were adjusted to have an equal concentration of total protein and subjected to PAAG electrophoresis followed by immunoblotting. Primary antibodies were purchased from Cell Signaling Technology. Secondary antibodies were purchased from Jackson Immuno Research.

### 4.17. Statistical Analysis

Statistical analysis was performed on the R programming language. Data are expressed as mean ± SD. Student’s t-test and two-way analysis of variance (ANOVA) were used to evaluate the statistical significance of the difference in two or more groups, respectively. A *p*-value less than 0.05 was considered significant.

## 5. Conclusions

Here, we tested the biological and medicinal activities of SSRE based on the effects of an extract on the transcriptome of mammalian cells. We enhanced the computational analysis of the data by integrating GSEA and CMAP analysis, which allowed us to define with high confidence biochemical pathways regulated by this plant extract and to predict its potential biological activities. This study shed light on the mechanisms of known anti-cancer activities of the extract and uncovered novel and unexpected anti-viral activity in the stimulation of autophagy. It also uncovered several limitations of using transcriptome analysis to predict the biological and medicinal actions of plants.

## Figures and Tables

**Figure 1 plants-11-02195-f001:**
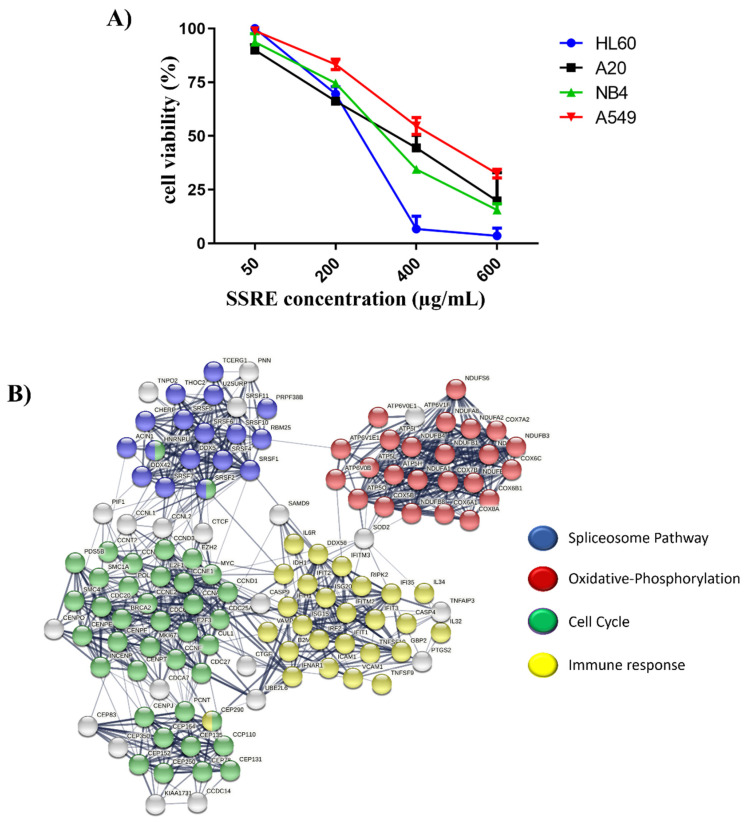
**Pathway analysis of Transcriptional effect of *SSRE* extract.** (**A**) Cytotoxicity assay of *Sarcopoterium spinosum* L. with different cell lines. A20 mouse lymphoma (black), HL-60 human promyelocytic leukemia (blue), NB4 cells (green), and A549 human lung cancer cells (red) were incubated with a range of concentrations of SSRE for 24 h, and an XTT assay was performed. (**B**) Comparative Gene Set Enrichment Analysis (GSEA) of extract effect on lung (A549) and Mouse Adipocyte (3T3) cell lines. In the middle, the number of common pathways that have been found in both cell lines after extract treatment has been described. (**C**) RNAseq-based protein-protein interaction networks of hub genes obtained from the STRING Database (interaction score > 0.9). (Isolated protein nodes were not depicted in Figure). The Intensity of nodes between genes represents the strength of the nodes and the interaction between proteins.

**Figure 2 plants-11-02195-f002:**
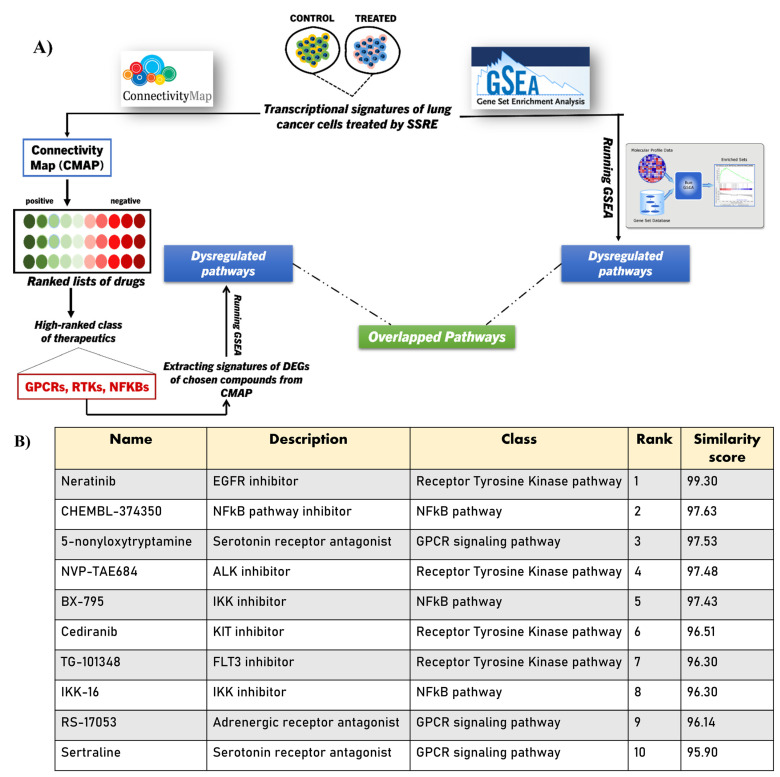
**Integration analysis of GSEA-CMAP.** (**A**) Graphical summary of the GSEA-CMAP integration analysis. (**B**) Top-ranked 10 CMAP compounds that induce transcriptional alterations similar to (indicated by positive similarity score) SSRE treatment of A549 cells. (**C**) Venn Diagram describing the overlapped pathways of three classes of compounds (RTKs, NFKBs, GPCRs).

**Figure 3 plants-11-02195-f003:**
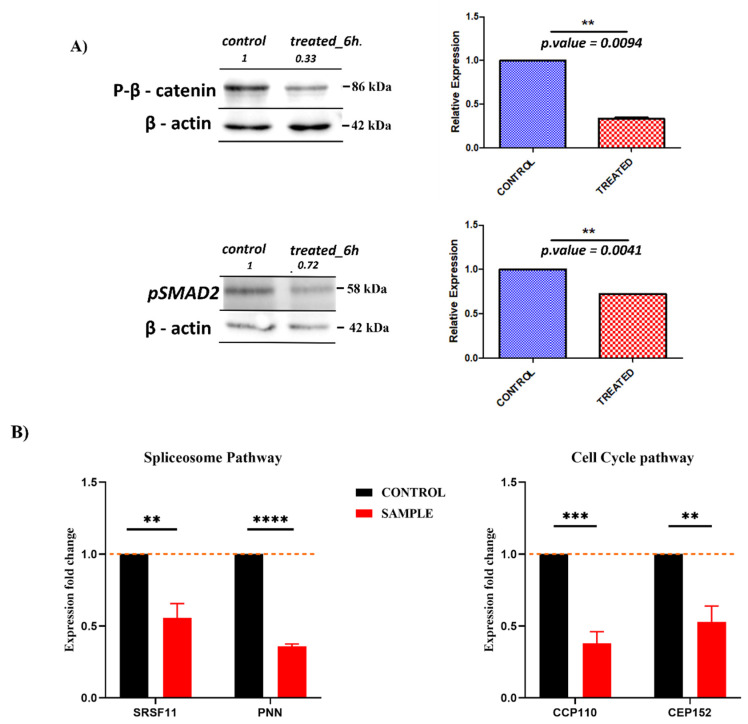
**Anti-cancer effects of the SSE extract.** (**A**) Effects of extract on WNT/Beta-catenin and TGF-beta signaling pathways. A549 cells were incubated with the extract for 6 h and levels of β-catenin and TGF-beta were assessed by immunoblotting with the corresponding antibodies. (**A**) Levels of β-catenin and level of SMAD2. (**B**) Real-Time PCR measurement of cancer-related pathways’ genes. On the left, Spliceosome genes (SRSF11 and PNN), and on the right genes belong to the Cell cycle pathway described (CCP110 and CEP152). Results are representative of three biological replicates (*n* = 3). *, *p* < 0.05; **, *p* < 0.01; ***, *p* < 0.005; ****, *p* < 0.001 (treated vs. control). (**C**) Cell cycle analysis. Flow Cytometry analysis of non-treated (on the left) and extract-treated (for 24 h) (on the right) A549 lung cancer cells. The Bar graph below shows the quantitative analysis of the flow cytometry experiments described above (*n* = 3). Data are presented as mean ± S.D. *, *p* < 0.05.

**Figure 4 plants-11-02195-f004:**
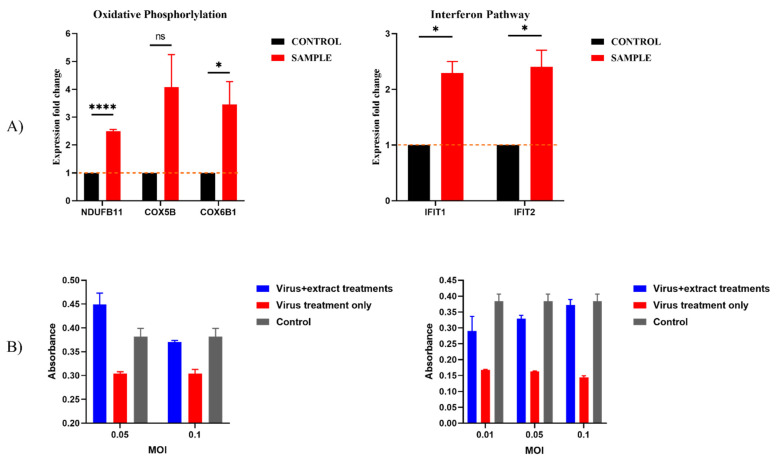
***Sarcopoterium spinosum* L. extract activates the anti-viral response.** (**A**) Real-Time PCR measurement of mitochondrial (NDUFB11, COX5B, COX6B1) and γ-interferon-induced genes (IFIT1, IFIT2). *, *p* < 0.05; **** *p* < 0.001 (treated (sample) vs. control). (**B**) The extract protects from the SARS-CoV-2 infection. Cells were pretreated with extract for 4 h, then SARS-CoV-2 virus was added to the cells, cell viability was measured by the MTT assay after 48 h. On the right viability of VERO6 cells, on the left viability of A549 cells have been described. Black—untreated control, Red—virus treatment only, Blue—virus + extract treatments.

**Figure 5 plants-11-02195-f005:**
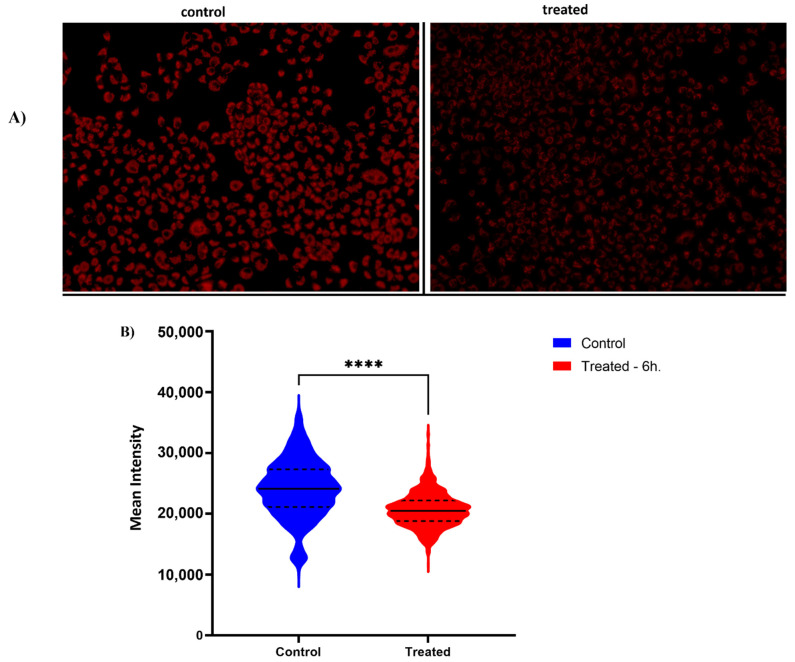
**Measurement of mitochondrial content in A549 cells.** A549 cells were treated with SSE (200 μg/mL) for 6 h. Fluorescence intensities of Mito-Tracker Red were quantified. Treatment with SSE reduces the number of mitochondria in cells. (**A**). Representative immunofluorescence images. The intensity of the fluorescent dye in each cell represents the amount of mitochondrial mass. Lower intensity of fluorescence MitoRed dye in treated cells compared to untreated cells demonstrates reduced mitochondria content (**B**). Violin plot demonstrates the difference in mitochondrial content between untreated (blue) and treated (red) samples. On the *Y*-axis Mean Intensity of the mitochondrial mass of the cells has been shown. Low intensity indicates fewer mitochondria inside the cells. ****, *p* < 0.001.

**Figure 6 plants-11-02195-f006:**
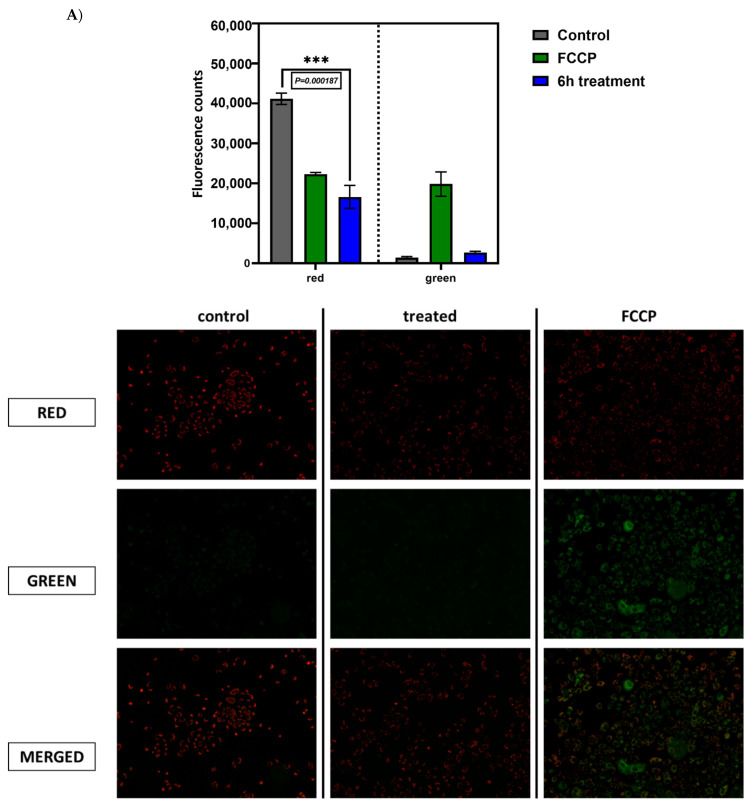
**Compensatory activity of SSRE through oxidative-phosphorylation pathway.** (**A**) **Mitochondrial membrane potential assay**. A549 cells were treated with 200 μg/mL extract for 6 h and stained with JC-1. Changes in mitochondrial membrane potential were detected using a JC-1 assay kit containing the cationic dye, which undergoes a readily detectable shift from red to green with decreases in membrane potential. The left panel of the bar graph (red—number of healthy mitochondria) describes a degradation of mitochondria after treatment with the extract. On the right panel (green—the number of unhealthy mitochondria), the level of damaging of the membrane potential of mitochondria after administration of extract was shown. Microscopy images demonstrate the number of mitochondria inside the cells in red, unhealthy mitochondria shown as green in A549 cells before and after exposure to SSE extract. FCCP (mitochondrial oxidative phosphorylation uncoupler) was used as a positive control which damages mitochondrial membrane potential. (**B**) **Mitochondrial autophagic degradation assay.** Representative immunofluorescence images using a confocal microscope. A549 were treated with 200 μg/mL extract and/or with autophagy inhibitor hcq (blocker of the lysosomal H^+^-ATPases hydroxychloroquine) with or without SSE administration for 6 h and stained with JC-1. The intensity of fluorescent dye represents the amount of mitochondrial content inside the cells. The bar graph indicates the statistical difference in the mitochondrial content of A549 cells. Imaging results are representative of three biological replicates (*n* = 3). ***, *p* < 0.005, *, *p* < 0.05.

**Figure 7 plants-11-02195-f007:**
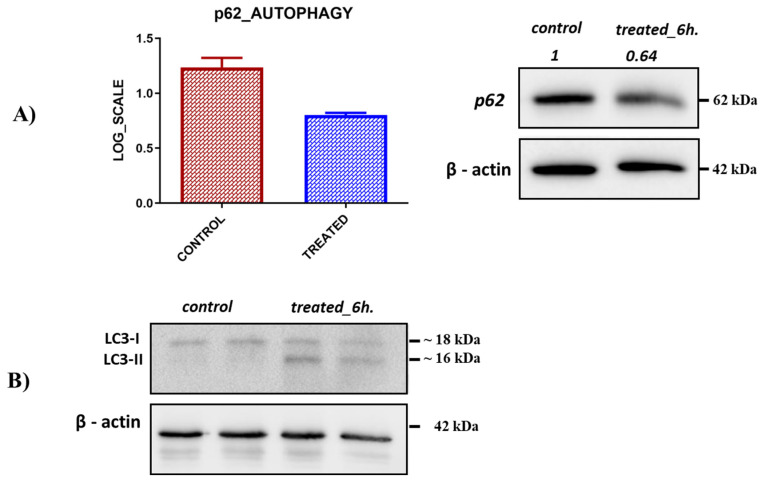
**Activation of autophagic pathway.** Effect of autophagy on p62 and LC3A/B protein level. (**A**) A decrease in the p62 protein level after treatment with the plant extract. (**B**) Increased level of LC3A/B following treatment with the extract. Cells were either treated or untreated with the extract for 6 h.

## Data Availability

The list of genes from transcriptome analysis and biological pathway analysis results with their corresponding raw and processed statistical values are available in the supplement section of this paper (Appendix A). All other necessary data is included in the main section of the paper. Codes and pipelines will be made available upon request.

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
