# Peer review of "Integration of the Connectivity Map and Pathway Analysis to Predict Plant Extract’s Medicinal Properties—The Study Case of Sarcopoterium spinosum L."

_plants, 2022, doi:10.3390/plants11172195_

Round 1
Reviewer 1 Report
The manuscript, "Integration of the Connectivity Map and pathway analysis to predict plant's extract medicinal properties-the study case of Sarcopoterium spinosum" presents a novel study for knowledge of the biological activities of medicinal plants. According to the results of both experiments and the use of bioinformatics data, the Authors suggest that the use of Connectivity Maps (CMAP) and gene set enrichment analysis (GSEA) can be used to predict biological activities of extracts, as well as evidence of biological activities at molecular level. The manuscript is quite interesting, and the results expressed in figures, tables and paragraphs make it clear for reading. However, the manuscript has some spelling errors that should be checked, for example the word "signaling" (signalling) and others. Likewise, I suggest that the main phytochemical components of the S. spinosum extract be mentioned in the introduction, as well as the effect that each group of main components could have on the biological activities reported in the manuscript be connected in the discussion.

Author Response
Reviewer 1:
The manuscript, "Integration of the Connectivity Map and pathway analysis to predict plant's extract medicinal properties-the study case of Sarcopoterium spinosum" presents a novel study for knowledge of the biological activities of medicinal plants. According to the results of both experiments and the use of bioinformatics data, the Authors suggest that the use of Connectivity Maps (CMAP) and gene set enrichment analysis (GSEA) can be used to predict the biological activities of extracts, as well as evidence of biological activities at the molecular level. The manuscript is quite interesting, and the results expressed in figures, tables, and paragraphs make it clear for reading. However, the manuscript has some spelling errors that should be checked, for example, the word "signaling" (signaling and others. Likewise, I suggest that the main phytochemical components of the S. spinosum extract be mentioned in the introduction, as well as the effect that each group of main components could have on the biological activities reported in the manuscript be connected in the discussion.
Answer: We appreciate the reviewer`s comments. After re-checking the whole manuscript, the English language was improved, and misspelled words were corrected. Regarding the phytochemical components of SSRE, we have added more information in the Introduction (lines 116-120) with corresponding references. Furthermore, in the Discussion (lines 332-339) we have discussed known activities of bioactive compounds that have been so far identified in the extract.

Reviewer 2 Report
Dear authors,After reviewing the following manuscript entitled "Integration of the conncectivity map and pathway analysis to predict plant extract’s medicinal properties – the study case of Sarcopoterium spinsosum" and with reference number (Plants - 1850363), I sent the following comments and observations that the authors should attend to before its publication in this journal.
I appreciate the work being well organized, with the experimental data explained. However, I have a few remarks:- To be specified if the plant has been identified on botanical databases. It should also be specified when the root was harvested and how it was prepared – was it used immediately after harvesting or was it dried.
- It should have been checked which extraction technique of bioactive compounds is the optimal one. How do the authors justify the extraction ratio used. Usually aqueous extracts are difficult to store. How was the solution preserved?
- It is not clear from the paper which bioactive compounds were extracted. Who is responsible for the biological activity. In such a study it must be specified which compounds are responsible for the pharmacological properties.
- I recommend using a reference substance. In this way the results would be more conclusive.
- Figures 5A are not clear
- To be corrected throughout the work mL by mL.
- The images in figure 6 are not clear
- Throughout the work, the binominal name of the plant must be written correctly - Sarcopoterium spinosum L.
- In figure 1, correct the data for concentration - µg/mL. Also specify which concentration it is about.
- The references older than 2010 should be removed. I am convinced that the authors can document themselves about other references.
Author Response
Reviewer 2:
After reviewing the following manuscript entitled "Integration of the connectivity map and pathway analysis to predict plant extract’s medicinal properties – the study case of Sarcopoterium spinosum" and with reference number (Plants - 1850363), I sent the following comments and observations that the authors should attend to before its publication in this journal.
I appreciate the work being well organized, with the experimental data explained. However, I have a few remarks:
- To be specified if the plant has been identified on botanical databases. It should also be specified when the root was harvested and how it was prepared – was it used immediately after harvesting or was it dried?
Author Answer: a). A voucher specimen of the plant was deposited in the Israel National Herbarium at the Hebrew University of Jerusalem (No. HUJ 102531). Now added to the text (line 384-385).
- b) The extract was prepared immediately after harvesting.
- It should have been checked which extraction technique of bioactive compounds is the optimal one. How do the authors justify the extraction ratio used? Usually, aqueous extracts are difficult to store. How was the solution preserved?
Author Answer: As we wanted to follow the ethnobotanical uses of this plant, we only tested the aqueous extract. The extract was lyophilized and kept at -200C as dry material. A working solution was freshly prepared at each time (see updated Materials and Methods).
- It is not clear from the paper which bioactive compounds were extracted. Who is responsible for the biological activity? In such a study it must be specified which compounds are responsible for the pharmacological properties.
Author Answer: The Aim of this study was to predict previously unknown activities of a plant extract. As a case test, we looked for new activities of a known medicinal plant SSRE, using transcriptomic changes. This integrational transcriptomic analysis should give us an initial indication of whether the plant has any activity at all and then decide whether to look for the active substances in it. Therefore, we focused on identifying novel activities in the whole extract and have not proceeded towards the isolation of novel active compounds.
Some known bioactive compounds from SSRE are addressed in the Introduction (line116-120) and Discussion (line 332-339).
- I recommend using a reference substance. In this way, the results would be more conclusive.
Author Answer: In the paper, we were validating a wide range of biochemical activities of SSRE. We used a reference substance FCCP in experiments with mitochondria to make sure whether the extract damages mitochondria. With other pathways, the extract had clear stimulating or inhibiting activities and we do not see what additional information could be gained by adding experiments with reference substances.
- Figures 5A are not clear
Author Answer: We modified the figure legend and improved it for better clarity.
- To be corrected throughout the work ml by mL.
Author Answer: Considering the comment, we have rechecked all volume measures (mL) and wrote them in the standardized style.
- The images in figure 6 are not clear
Author Answer: We agree with the reviewer`s comment. For a better explanation, we improved the resolution of the microscopy pictures in figure 6 and improved the visualization of the figure. Additionally, figure legends are modified for a better understanding.
- Throughout the work, the binominal name of the plant must be written correctly - Sarcopoterium spinosum L.
Author Answer: Binominal Latin name of the plant was rechecked throughout the whole manuscript and written correctly.
- In figure 1, correct the data for concentration - µg/mL. Also, specify which concentration it is about.
Author Answer: We agree with the comment on correction in figure 1 which demonstrates the cytotoxicity effect of plant extract in different cancer cell lines. Representation of concentration unit corrected to µg/mL version.
- The references older than 2010 should be removed. I am convinced that the authors can document themselves about other references.
Author Answer: We have cleared out some of the references that are older than 2010. However, there are some articles that are really crucial to mention in terms of the traditional usage of the plant and it is initial medicinal activity.

Round 2
Reviewer 2 Report
I think it can be published.